# Comparison of different techniques for prehospital cervical spine immobilization: Biomechanical measurements with a wireless motion capture system

**Sarah Morag**[1], **Martin Kieninger**[1]*, **Christoph Eissnert**[1], **Simon Auer**[2,3], **Sebastian Dendorfer**[2,3], **Daniel Popp**[4], **Johannes Hoffmann**[5], **Bärbel Kieninger**[5]

1 Department of Anesthesiology, University Medical Center Regensburg, Regensburg, Germany,
2 Laboratory for Biomechanics, Ostbayerische Technische Hochschule (OTH) Regensburg, Regensburg, Germany, 3 Regensburg Center for Biomedical Engineering, Ostbayerische Technische Hochschule (OTH) Regensburg, Regensburg, Germany, 4 Department of Trauma Surgery, University Medical Center Regensburg, Regensburg, Germany, 5 Department of Infection Prevention and Infectious Diseases, University Medical Center Regensburg, Regensburg, Germany

* martin.kieninger@ukr.de

**Data Availability Statement:** All relevant data are within the paper and its Supporting Information files.

## Abstract

### Background

Various rescue techniques are used for the prehospital transport of trauma patients. This study compares different techniques in terms of immobilization of the cervical spine and the rescue time.

### Methods

A wireless motion capture system (Xsens Technologies, Enschede, The Netherlands) was used to record motion in three-dimensional space and the rescue time in a standardized environment. Immobilization was performed by applying different techniques through different teams of trained paramedics and physicians. All tests were performed on the set course, starting with the test person lying on the floor and ending with the test person lying on an ambulance cot ready to be loaded into an ambulance. Six different settings for rescue techniques were examined: rescue sheet with/without rigid cervical collar (P1S1, P1S0), vacuum mattress and scoop stretcher with/without rigid cervical collar (P2S1, P2S0), and long spinal board with/without rigid cervical collar (P3S1, P3S0). Four time intervals were defined: the time interval in which the rigid cervical collar is applied (T0), the time interval in which the test person is positioned on rescue sheet, vacuum mattress and scoop stretcher, or long spinal board (T1), the time interval in which the test person is carried to the ambulance cot (T2), and the time interval in which the ambulance cot is rolled to the ambulance (T3). An ANOVA was performed to compare the different techniques.

**Funding:** The authors received no specific funding for this work.

**Competing interests:** Martin Kieninger serves as academic editor for PLOS ONE. This does not alter our adherence to PLOS ONE policies on sharing data and materials.

## Results

During the simulated extrication procedures, a rigid cervical collar provided biomechanical stability at all angles with hardly any loss of time (mean angle ranges during T1: axial rotation P1S0 vs P1S1 p<0.0001, P2S0 vs P2S1 p<0.0001, P3S0 vs P3S1 p<0.0001; lateral bending P1S0 vs P1S1 p = 0.0263, P2S0 vs P2S1 p<0.0001, P3S0 vs P3S1 p<0.0001; flexion/extension P1S0 vs P1S1 p = 0.0023, P2S0 vs P2S1 p<0.0001). Of the three techniques examined, the scoop stretcher and vacuum mattress were best for reducing lateral bending in the frontal plane (mean angle ranges during T1: P2S1 vs P3S1 p = 0.0333; P2S0 vs P3S0 p = 0.0123) as well as flexion and extension in the sagittal plane (mean angle ranges during T2: P1S1 vs P2S1 p<0.0001; P1S0 vs P2S0 p<0.0001). On the other hand, the rescue sheet was clearly superior in terms of time (total duration P1S0 vs P2S0 p<0.001, P1S1 vs P2S1 p<0.001, P1S0 vs P3S0 p<0.001, P1S1 vs P3S1 p<0.001) but was always associated with significantly larger angular ranges of the cervical spine during the procedure. Therefore, the choice of technique depends on various factors such as the rescue time, the available personnel, as well as the severity of the suspected instability.

## Background

Major injuries are often associated with spinal trauma, which carries the risk of not being recognized in time or not being recognized at all. According to a Europe-wide study based on data from the Trauma Audit and Research Network, 13% of patients who sustain major trauma are also affected by spinal trauma, in 45% of cases of the cervical spine [1]. In Germany, rates of spinal involvement range between 17% and 34% [2, 3].

Various techniques are used for the prehospital rescue of trauma patients, and the choice of techniques is often influenced by its international usage. For example, the standard treatment protocol of the United States stipulates that every patient with certain trauma mechanisms–regardless to the actual injury pattern–has to unconditionally undergo complete spinal immobilization with a rigid cervical collar and a long spinal board [4], which has led to unnecessary waste of time, personnel, and diagnostic resources [5].

Therefore, current algorithms and guidelines vary among countries, and authorities aim to incorporate new findings into their decision-making in prehospital immobilization processes, for instance, by taking into account clinical findings such as consciousness and neurological deficits [5]. In this context, it is worse to mention the application of the nexus criteria for or against the application of a rigid cervical collar in Norwegian guidelines for the prehospital management of adult trauma patients [6]. A research group in the United Kingdom has recently also looked at similar trade-offs, for example, spinal motion restriction vs rigid immobilization, and has called for a change of the still rather narrow preclinical guidelines [7]. Yet, the guideline for the treatment of polytrauma and severe injuries issued by the German Trauma Society in December 2022 clearly recommends stabilization of the cervical spine prior to the actual technical rescue, except in cases in which patients require immediate rescue, such as in fires or risk of explosion. Nevertheless, it is emphasized that there is no true evidence of a positive effect of stabilization [8].

Consequently, the generation of robust biomechanical data on the optimal approach to cervical spine immobilization seems to be an important issue with regard to the development of future guidelines.

## Material and methods

### Aim of the study

This study compared different techniques of spinal immobilization used during the prehospital transport of trauma patients. The time period from the beginning of the patient's recovery to the loading into the ambulance is considered. The focus was on the time required for performing the extraction procedure and the range of motion of the cervical spine in terms of its axial rotation, lateral bending, as well as flexion or extension during the procedure.

### Ethics approval and consent to participate

The study was approved by and conducted according to the guidelines of the Ethics Committee of the University of Regensburg (approval number 20-1661-101). All study participants consented in writing to their participation in the study and the use of the photographs taken. Participants were recruited in October 2020.

### Study design

This study was planned as an explorative analysis of biomechanical aspects of prehospital immobilizing processes. The techniques compared are summarized in Table 1.

Spinal motion parameters were measured by immobilizing a test person in a standardized manner using one of the various techniques commonly used by German rescue services. Emergency service personnel were organized in 11 teams of two. In total, 19 persons (1 physician, 12 active rescue professionals (paramedics and emergency medical technical), and 6 trained volunteer medical personnel) performed immobilization in a state-of-the-art manner. In a team of two, one person fixed the head while the other performed the other procedures. Two healthy male volunteers (subject 1: age 22, height 179cm, weight 73kg; subject 2: age 26, height 181cm, weight 96kg) were available as test persons. The setup of the study is shown in Fig 1. The structure of the temporal sequence was divided into structural time intervals by defined time markers (Table 2).

**Table 1. Description of the three different techniques, called P1, P2, and P3, each performed with (S1) and without (S0) a rigid cervical collar.**

| Name technique | Description |
|---|---|
| **P1S0** | Rescue sheet technique without a rigid cervical collar |
| | (Rescue sheet: "Rettungstuch" Söhngen) |
| **P1S1** | Rescue sheet technique with a rigid cervical collar |
| | (Rigid cervical collar: Ambu perfit ACE, Ambu Rescue sheet: "Rettungstuch", Söhngen) |
| **P2S0** | Vacuum mattress technique including scoop stretcher transfer without a rigid cervical collar |
| | (Scoop stretcher: „Schaufeltrage", Söhngen Vacuum mattress: "Vakuummatratze", Schnitzler) |
| **P2S1** | Vacuum mattress technique including scoop stretcher transfer with a rigid cervical collar |
| | (Rigid cervical collar: Ambu perfit ACE, Ambu Scoop stretcher: „Schaufeltrage", Söhngen Vacuum mattress: "Vakuummatratze", Schnitzler) |
| **P3S0** | Long spinal board technique without a rigid cervical collar |
| | (Long spinal board, Headblocks, Spider Straps: Baxstrap, Laerdal) |
| **P3S1** | Long spinal board technique with a rigid cervical collar |
| | (Rigid cervical collar: Ambu perfit ACE, Ambu Long spinal board, Headblocks, Spider Straps: Baxstrap, Laerdal) |

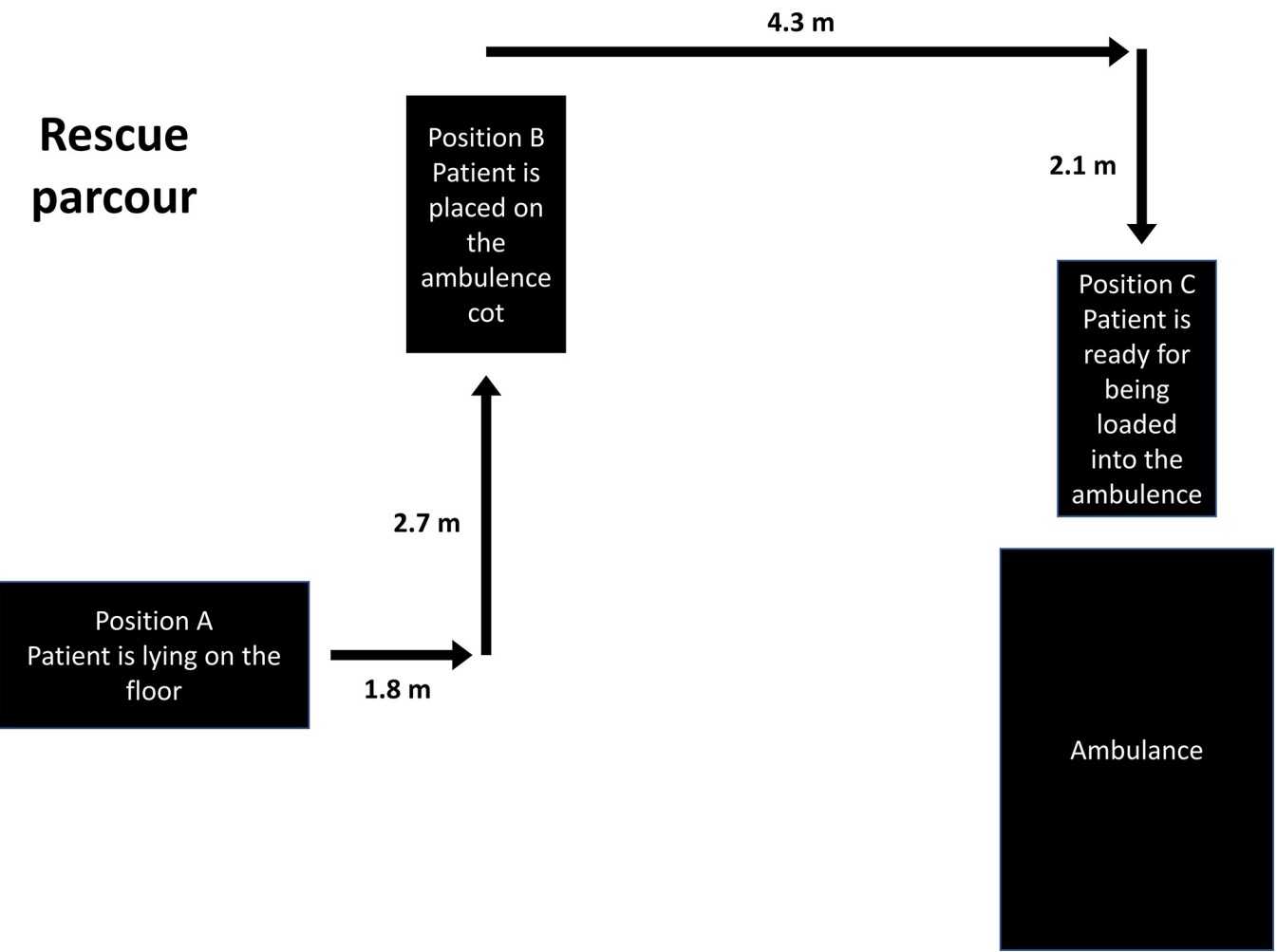

**Fig 1. Trial setup.** Position A–Starting position with test person lying down (immobilization by means of rescue sheet and the long spinal board ends here); Position B—Interstage 1 with ambulance cot lowered, ready for the movement of the test person (immobilization by means of vacuum mattress ends here); Position C–Interstage 2 with ambulance cot elevated, ready for loading the test person into the ambulance.

**Table 2. Time sequence of a test run, defined by the time markers.** Note that the interval T0 and the corresponding marker M0 are omitted for test runs without the application of a rigid cervical collar.

| Time interval | Marker | Description |
|---|---|---|
| | Start | Start of measurement: test person in position A |
| T0 | | Application of a rigid cervical collar |
| | M0 | Fully applied rigid cervical collar with test person in position A |
| T1 | | Positioning on rescue sheet, vacuum mattress or spinal board |
| | M1 | Test person lying on rescue sheet, vacuum mattress or spinal board in position A |
| T2 | | Test person carried from position A to position B |
| | M2 | Test person lying on the ambulance cot in position B |
| T3 | | Ambulance cot rolled to the ambulance |
| | M3 | End of measurement: Ambulance cot is placed in position C immediately in front of the ambulance |

## Data collection

For data acquisition, the Xsens MVN measurement system (Xsens Technologies, Enschede, The Netherlands) was used. For this purpose, a motion capture suit with several sensors was attached to the body of the test person. By means of body parameters initially taken for system calibration, the measurement started with the test person lying in a physiologically neutral resting position in position A (Fig 1). The start of the measurements from this position was subsequently optimized by optical adjustment in the Xsens MVN software during the analysis phase.

A total of 54 measurements (9 runs of each setting) were performed with the above-mentioned techniques on two consecutive days. The MVN system was regularly recalibrated during the ongoing measurements taken in a standardized manner and delivered final motion data with a frequency of 240 Hz.

## Statistical analysis

The raw data from the measurements were imported into the Xsens MVN software (version 2021.2.0) and transferred to the underlying biomechanical model (multi-level); the coordinates and angles calculated in the process were output as an Excel file. For movements of the cervical spine, the angle termed ergonomic joint angle Head_T8 in Xsens was considered, which three-dimensionally captures the rotational movements between the head and the sternum as Euler angle (representation of the angle ZXY with Z craniocaudal axis, X sagittal axis, and Y horizontal axis).

Then a transformation of the solid angles was performed so that the angle measured at the starting position was the zero angle, and all further movements were considered with respect to this angle. The measured values were smoothened with a frequency of 5 Hz. The mean values, their standard deviations, and the maximum and minimum values for each of the 54 measurements were calculated for all measurements as well as for the individual subsections of the experimental setup. The spatial angle transformation, the smoothing, and the calculation of the descriptive angles were calculated with Python 3.9 (Anaconda 2021.11, Ubuntu 21.10, python code S1 File). For a comparison of the six different experimental settings, the values (mean maximum absolute angle, mean maximum angle, and mean minimum angle) were compared using a three-way (technique, use of rigid cervical collar, time interval) ANOVA (SAF, version 9.4). A Mann-Whitney U test (2-sided, significance level 0.05) was used to compare the total times of the test runs (SPSS Statistics, version 28.0.1.0).

## Results

### Duration analysis

For each technique, the mean duration and its standard deviation, calculated from the nine tests performed per technique, were determined for each of the time periods T0, T1, T2, and T3 (Table 3). For each case, the mean duration of the complete runs Ttotal (T0+T1+T2+T3 for tests with a rigid cervical collar and T1+T2+T3 for tests without a rigid cervical collar), their standard deviation, as well as the minimum and maximum are also summarized in Table 3. The significantly shorter time Ttotal of the rescue sheet technique compared to that of the other two techniques is striking. The application of the rigid cervical collar (T0) took a mean of 22 ± 8s.

The p-values calculated in the total duration comparison are shown in S1 Table. The duration of the rescue sheet technique was significantly shorter than those of the vacuum mattress

**Table 3. The mean duration of the intervals T0, T1, T2, and T3, its standard deviation, as well as the minimum and maximum for T0, T1, T2, and T3 related to all test runs of one technique were calculated from the nine trials per test setup and are given in seconds.**

|  | T0 | T1 | T2 | T3 | Ttotal |
|---|---|---|---|---|---|
| **P1S0** | - | 48 ± 17s (29s – 76s) | 94 ± 20s (70s – 127s) | 19 ± 7s (12s – 36s) | 185 ± 27s (150s – 233s) |
| **P1S1** | 21 ± 7s (13s – 37s) | 50 ± 15s (33s – 77s) | 72 ± 12s (56s – 90s) | 16 ± 4s (10s – 28s) | 186 ± 39s (130s – 247s) |
| **P2S0** |  | 214 ± 69s (146s – 335s) | 56 ± 17s (33s – 83s) | 18 ± 5s (11s – 26s) | 315 ± 83s (236s – 440s) |
| **P2S1** | 23 ± 8s (15s – 38s) | 209 ± 69s (116s – 330s) | 57 ± 27s (24s – 110s) | 20 ± 5s (12s – 29s) | 327 ± 85s (226s – 466s) |
| **P3S0** | - | 257 ± 54s (182s – 323s) | 83 ± 26s (55s – 114s) | 13 ± 2s (9s – 15s) | 372 ± 63s (279s – 456s) |
| **P3S1** | 23 ± 10s (8s – 41s) | 246 ± 78s (147s – 383s) | 76 ± 19s (54s – 105s) | 18 ± 7s (11s – 31s) | 402 ± 88s (283s – 517s) |

technique and the long spinal board technique (both with and without a rigid cervical collar: p<0.001).

## Analysis of axial rotation around the cranio-caudal body axis

For each test setup, the mean value of the absolute angles per time interval (for reasons of symmetry, no distinction was made between left and right rotation of the axis) and its standard deviation of axis rotation were calculated; likewise, the mean maximum of this absolute angle was determined and is summarized in Table 4. The range of motion that was swept during the respective time interval (angle range: maximum value in positive direction minus maximum value in negative direction) was calculated for each case.

In interval T1, both the mean maximum angle and the mean angle range of axial rotation were significantly smaller in the test setup with than in the setup without a rigid cervical collar (mean maximum absolute angle: P1S0 vs P1S1 $p = <0.0001$; P2S0 vs P2S1 $p = 0.0030$; P3S0 vs P3S1 $p = <0.0001$; mean angle range: P1S0 vs P1S1 $p = <0.0001$; P2S0 vs P2S1 $p = <0.0001$; P3S0 vs P3S1 $p<0.0001$). Meanwhile, in interval T2, the mean angle range of rotation were significantly larger in the setup with a rescue sheet than in the setups with a vacuum mattress and

**Table 4. Mean absolute angle, mean maximum absolute angle, and mean angle range with standard deviation, each given in degrees.**

|  |  | T0 | T1 | T2 | T3 |
|---|---|---|---|---|---|
| **mean absolute angle** | **P1S0** | - | 6 ± 2° | 5 ± 4° | 5 ± 4° |
|  | **P1S1** | 3 ± 2° | 4 ± 1° | 4 ± 2° | 4 ± 2° |
|  | **P2S0** | - | 3 ± 1° | 4 ± 2° | 3 ± 2° |
|  | **P2S1** | 2 ± 1° | 2 ± 1° | 2 ± 1° | 2 ± 1° |
|  | **P3S0** | - | 6 ± 7° | 5 ± 9° | 5 ± 9° |
|  | **P3S1** | 2 ± 2° | 4 ± 3° | 4 ± 3° | 4 ± 3° |
| **mean maximum absolute angle** | **P1S0** | - | 20 ± 6° | 15 ± 7° | 8 ± 4° |
|  | **P1S1** | 6 ± 3° | 8 ± 3° | 9 ± 3° | 5 ± 2° |
|  | **P2S0** | - | 14 ± 9° | 5 ± 2° | 4 ± 3° |
|  | **P2S1** | 6 ± 2° | 5 ± 2° | 4 ± 1° | 3 ± 1° |
|  | **P3S0** | - | 27 ± 16° | 7 ± 8° | 6 ± 9° |
|  | **P3S1** | 5 ± 3° | 10 ± 6° | 5 ± 3° | 4 ± 3° |
| **mean angle range** | **P1S0** | - | 28 ± 8° | 21 ± 10° | 5 ± 4° |
|  | **P1S1** | 8 ± 4° | 9 ± 3° | 11 ± 5° | 2 ± 1° |
|  | **P2S0** | - | 21 ± 12° | 5 ± 2° | 2 ± 1° |
|  | **P2S1** | 7 ± 5° | 7 ± 2° | 3 ± 2° | 1 ± 1° |
|  | **P3S0** | - | 32 ± 18° | 3 ± 1° | 1 ± 1° |
|  | **P3S1** | 5 ± 3° | 12 ± 10° | 3 ± 1° | 1 ± 1° |

a long spinal board (P1S0 vs P2S0 p<0.0001; P1S0 vs P3S0 p<0.0001; P1S1 vs P2S1 p = 0.0189; P1S1 vs P3S1 p = 0.0116). In the same interval, the mean maximum angle and the mean angle range of rotation were significantly larger in the experimental setup with a rescue sheet without a rigid cervical collar than in the same setup with a rigid cervical collar (mean maximum absolute angle: P1S0 vs P1S1 p = 0.0362; mean angle range: P1S0 vs P1S1 p = 0.0007). The results of the significance tests are summarized in S2 Table.

## Analysis of lateral bending in the frontal plane

Analogous to axial rotation, the angles of lateral bending were analyzed: For each experimental setup, the mean value of the absolute angle per time interval (again, no distinction was made between left and right flexion) and its standard deviation were calculated. Table 5 also shows the mean maximum of this absolute angle and the respective range of motion swept during the respective time interval (maximum value in positive direction minus maximum value in negative direction).

The results of the significance tests corresponding to Table 5 are recorded in S3 Table. In interval T1, the mean angle range in lateral bending in all setups with a rigid cervical collar was significantly smaller than in the same setups without a rigid cervical collar (P1S0 vs P1S1 p = 0.0263; P2S0 vs P2S1 p<0.0001; P3S0 vs P3S1 p<0.0001). In addition, the mean angle range is significantly smaller for the technique with vacuum mattress than for the long spinal board technique (P2S0 vs P3S0 p = 0.0123; P2S1 vs P3S1 p = 0.0333). Meanwhile, in interval T2, the mean angular range in lateral flexion was significantly larger for the rescue sheet technique than for the vacuum mattress and long spinal board techniques in all tests (P1S0 vs P2S0 p<0.0001; P1S0 vs P3S0 p<0.0001; P1S1 vs P2S1 p = 0.0024; P1S1 vs P3S1 p = 0.0105).

## Analysis of flexion and extension in the sagittal plane

For each test, the motions in flexion and extension were analyzed by means of the mean angle, the mean minimum angle, the mean maximum angle, and the mean angle range of motion including their standard deviation per time interval (Table 6).

**Table 5. Mean absolute angle, mean maximum absolute angle and mean angle range with standard deviation each given in degrees.**

|  |  | T0 | T1 | T2 | T3 |
|---|---|---|---|---|---|
| mean absolute angle | P1S0 | - | 5 ± 2˚ | 7 ± 5˚ | 7 ± 3˚ |
|  | P1S1 | 4 ± 3˚ | 6 ± 2˚ | 5 ± 3˚ | 6 ± 4˚ |
|  | P2S0 | - | 4 ± 2˚ | 7 ± 4˚ | 6 ± 4˚ |
|  | P2S1 | 2 ± 1˚ | 3 ± 2˚ | 4 ± 2˚ | 4 ± 3˚ |
|  | P3S0 | - | 4 ± 3˚ | 5 ± 5˚ | 6 ± 4˚ |
|  | P3S1 | 2 ± 1˚ | 3 ± 2˚ | 3 ± 3˚ | 4 ± 2˚ |
| mean maximum absolute angle | P1S0 | - | 13 ± 4˚ | 14 ± 10˚ | 11 ± 3˚ |
|  | P1S1 | 8 ± 4˚ | 11 ± 3˚ | 12 ± 6˚ | 8 ± 7˚ |
|  | P2S0 | - | 4 ± 4˚ | 8 ± 4˚ | 9 ± 6˚ |
|  | P2S1 | 6 ± 3˚ | 6 ± 2˚ | 6 ± 3˚ | 7 ± 4˚ |
|  | P3S0 | - | 17 ± 8˚ | 7 ± 4˚ | 8 ± 3˚ |
|  | P3S1 | 5 ± 3˚ | 9 ± 3˚ | 5 ± 3˚ | 6 ± 4˚ |
| mean angle range | P1S0 | - | 17 ± 5˚ | 17 ± 9˚ | 7 ± 5˚ |
|  | P1S1 | 8 ± 4˚ | 12 ± 5˚ | 11 ± 5˚ | 6 ± 7˚ |
|  | P2S0 | - | 17 ± 5˚ | 4 ± 2˚ | 5 ± 5˚ |
|  | P2S1 | 7 ± 2˚ | 7 ± 3˚ | 3 ± 2˚ | 5 ± 5˚ |
|  | P3S0 | - | 23 ± 10˚ | 4 ± 1˚ | 4 ± 4˚ |
|  | P3S1 | 6 ± 4˚ | 12 ± 4˚ | 4 ± 2˚ | 5 ± 4˚ |

**Table 6. Mean angle, mean minimum angle, mean maximum angle, mean angle range with standard deviation each, given in degrees.** Angle values in a negative range describe extension, angle values in a positive range indicate flexion of the cervical spine.

|  |  | T0 | T1 | T2 | T3 |
|---|---|---|---|---|---|
| **mean angle** | P1S0 | - | 0 ± 5˚ | 2 ± 6˚ | -5 ± 6˚ |
|  | P1S1 | -3 ± 12˚ | -3 ± 16˚ | -4 ± 15˚ | -9 ± 12˚ |
|  | P2S0 | - | -8 ± 4˚ | -16 ± 5˚ | -16 ± 4˚ |
|  | P2S1 | -2 ± 6˚ | -12 ± 8˚ | -15 ± 7˚ | -15 ± 7˚ |
|  | P3S0 | - | 2 ± 2˚ | 0 ± 3˚ | -2 ± 2˚ |
|  | P3S1 | -3 ± 8˚ | -4 ± 13˚ | -6 ± 21˚ | -7 ± 11˚ |
| **mean minimum angle** | P1S0 | - | -11 ± 7˚ | -11 ± 5˚ | -7 ± 6˚ |
|  | P1S1 | -10 ± 13˚ | -9 ± 14˚ | -10 ± 12˚ | -10 ± 12˚ |
|  | P2S0 | - | -19 ± 7˚ | -19 ± 5˚ | -17 ± 5˚ |
|  | P2S1 | -8 ± 10˚ | -18 ± 8˚ | -17 ± 6˚ | -16 ± 7˚ |
|  | P3S0 | - | -11 ± 5˚ | -3 ± 3˚ | -2 ± 2˚ |
|  | P3S1 | -7 ± 11˚ | -12 ± 12˚ | -9 ± 10˚ | -8 ± 10˚ |
| **mean maximum angle** | P1S0 | - | 16 ± 11˚ | 40 ± 14˚ | -2 ± 7˚ |
|  | P1S1 | 5 ± 6˚ | 7 ± 21˚ | 21 ± 32˚ | 9 ± 12˚ |
|  | P2S0 | - | 9 ± 6˚ | -10 ± 7˚ | -14 ± 4˚ |
|  | P2S1 | 6 ± 4˚ | -4 ± 11˚ | -13 ± 7˚ | -16 ± 7˚ |
|  | P3S0 | - | 11 ± 4˚ | 4 ± 4˚ | -1 ± 3˚ |
|  | P3S1 | 4 ± 4˚ | 3 ± 15˚ | -4 ± 12˚ | -7 ± 11˚ |
| **mean angle range** | P1S0 | - | 26 ± 8˚ | 51 ± 13˚ | 5 ± 4˚ |
|  | P1S1 | 15 ± 11˚ | 16 ± 8˚ | 31 ± 22˚ | 1 ± 1˚ |
|  | P2S0 | - | 28 ± 6˚ | 9 ± 4˚ | 3 ± 2˚ |
|  | P2S1 | 14 ± 13˚ | 14 ± 7˚ | 4 ± 1˚ | 2 ± 1˚ |
|  | P3S0 | - | 22 ± 5˚ | 7 ± 2˚ | 2 ± 1˚ |
|  | P3S1 | 11 ± 9˚ | 15 ± 6˚ | 6 ± 2˚ | 1 ± 1˚ |

In interval T1, the mean angle range from extension to flexion was significantly smaller in each setup with a rigid cervical collar than in the same setups without a rigid cervical collar for the rescue sheet technique and the technique with vacuum mattress (P1S0 vs P1S1 p = 0.0023; P2S0 vs P2S1 p<0.0001). In interval T2, the mean minimum angle, the mean maximum angle, and the mean angle range from extension to flexion were significantly smaller in each setups without a rigid cervical collar and with the vacuum mattress and long spinal board techniques than with the rescue sheet technique (mean minimum angle: P1S0 vs P2S0 p = 0.0385; P1S0 vs P3S0 p = 0.0423; mean maximum angle: P1S0 vs P2S0 p<0.0001; P1S0 vs P3S0 p<0.0001; mean angle range: P1S0 vs P2S0 p<0.0001; P1S0 vs P3S0 p<0.0001). In the same interval, the mean maximum angle and the mean angle range were also significantly smaller in all tests with a rigid cervical collar in the setup with a vacuum mattress and a long spinal board than in the setup with a rescue sheet (mean maximum angle: P1S1 vs P2S1 p<0.0001; P1S1 vs P3S1 p<0.0001; mean angle range: P1S1 vs P2S1 p<0.0001; P1S1 vs P3S1 p<0.0001). These results are shown in tabular form in S4 Table.

## Graphical representation of the motion sequence

For each trial, the axial angles of rotation, the lateral angles, as well as the angles at flexion and extension were plotted against time and then compared (S1 Appendix). This method allowed investigating the aspect that the motions in the different spatial axes may not be considered separately but may sometimes occur simultaneously. One plot per experimental setup was selected as an example (Fig 2).

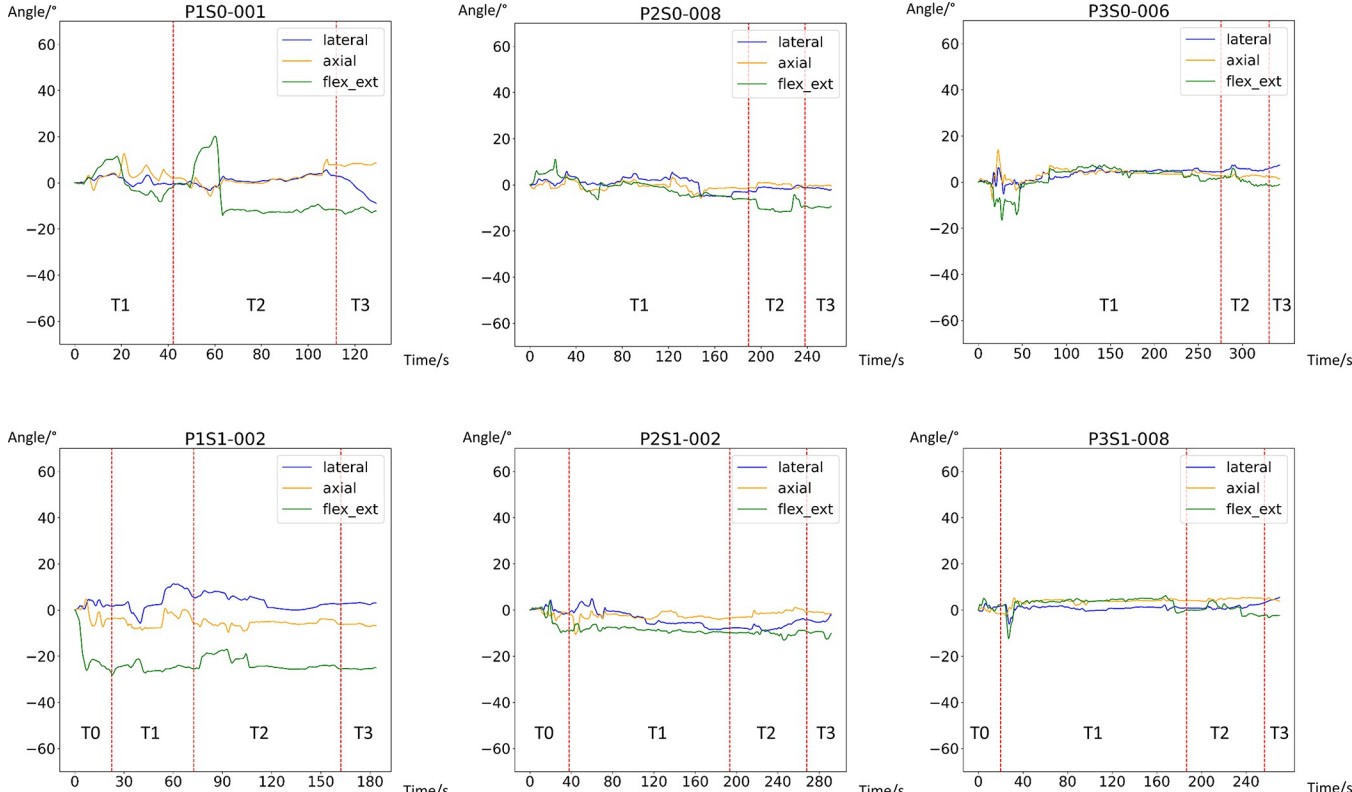

**Fig 2. Graphical representation of the three angles of movement (y axis) over the time intervals (x axis, time given in seconds).** Blue–lateral angle: negative value: right; positive: left; orange–axial rotation: negative: right; positive: left; green–extension/flexion: negative: extension; positive: flexion.

The following conspicuous features stand out: In the test setup with the rescue sheet without a rigid cervical collar, passive flexion of the cervical spine during the lifting procedure was over 20 degrees from interval T2 onwards; this flexion was maintained during the wearing period, followed by an abrupt extension movement when the test person was placed onto the stretcher. This degree of flexion is less pronounced with the use of a rigid cervical collar. In addition, a combination of axial rotation, lateral flexion, and especially extension of the cervical spine can be observed during manual head fixation, while performing the obligatory log-roll maneuver to transfer the person to the rescue sheet or the long spinal board. This movement was also present during the tests with a rigid cervical collar but less pronounced in terms of intensity and duration.

In addition, hyperextension of the cervical spine compared to the neutral position can be observed in all techniques after the application of a rigid cervical collar, which is usually maintained throughout the procedure. This peculiarity became particularly obvious when the plots of the two test persons were compared: the conspicuousness of hyperextension was visually much clearer in subject 1 than in subject 2 (S1 Appendix).

## Discussion

Previous studies have already addressed the issue of the range of motion of the cervical spine during certain emergency rescue procedures, relying on Xsens technology to record and subsequently analyze movements [9], as we have done in this study. However, in addition to other studies -for example, by Nolte et al. [10] who examined prehospital patient transport-, this

study focuses on the immobilization process itself, i.e., the active or passive movements that take place when putting a patient into a stabilized position.

Jung et al. [11] compared different models of rigid cervical collars. The Ambu Perfit ACE model used in our trials had been established as the model that allowed the least residual range of motion in comparison to the other models tested. Furthermore, use of this rigid cervical collar was shown to be an effective means of improving fixation in all our test setups, in particular by providing support to the cervical spine during log rolling by two paramedics. Moreover, this additional measure did not show any appreciable prolongation of the immobilization process.

However, several studies have pointed out possible side effects of a rigid neck brace, such as difficult airway management [12], cranial pressure due to drainage obstruction [13], or lack of patient compliance. Therefore, Phaly et Khan [14] recommended performing cervical spine immobilization only in patients who are considered particularly high-risk. Uzun et al. [15] showed that headblocks and various harness systems may also effectively reduce range of motion. In that study, however, measurements were started only after immobilization. In contrast, our study focuses on the process of immobilization itself, which is not only the first step in an emergency extrication but also the situation with the strongest movements.

Although no relevant difference in flexion and extension was found between vacuum mattresses and long spinal boards, use of a vacuum mattress in combination with a scoop stretcher seemed to be superior to use of a long spinal board in the investigated directions of axial rotation and lateral bending. Apparently, techniques requireing a log-roll maneuver of the patient cause increased movement in these directions, whether or not the cervical spine is stabilized manually or a rigid cervical collar is applied. Other trials have also found significant shifts between head and torso during log-rolling [16]. Because ambulances are manned by two persons by default, rotation of the patient has to be performed by only one person while the second person is holding the patient's head. This procedure explains the stronger axial rotation and lateral bending that was also be measured in our experiments. On top of finding, use of a vacuum mattress resulted in a slightly shorter immobilization procedure than use of a long spinal board.

Regarding the immobilization effect, the biomechanical study by Prasarn et al. also showed that use of a vacuum mattress was superior when moving cadavers with unstable subaxial injuries [17].

Furthermore, it should be taken into account that during transport long spinal boards cannot adapt as well as vacuum mattresses to the anatomically given kyphoses and lordoses of the spine, forcing the spine into an unnatural, straight posture during immobilization. During spinal instability, the pressure of the body weight on the straight surface creates tension within the spine, which may lead to displacement or further displacement at the unstable body site [18].

The long spinal board is often described as being uncomfortable for the patient [19] and as causing pressure ulcers if left in place for a long time [20]. Apart from the resulting tissue lesions, the uncomfortable positioning may divert the focus from actual injuries and lead to unnecessary radiological examinations with a corresponding delay in treatment [21]. Over all, another study with healthy participants found significantly decreased forced expiratory pressure in 1 second and forced vital capacity, and both decreased even more with extended immobilization time [22].

In cases of emergency, the following guiding objectives must be taken into account: If the timing of a rescue plays a decisive role, the rescue sheet clearly seems to be the goal-oriented method: In the course chosen in this study, which covered the rescue of an injured person up to the loading of the person into the rescue vehicle, rescue with the rescue sheet took only about half as long as rescue with the vacuum mattresses and the long spinal board techniques.

## Limitations

Our teams consisted of one very experienced person with a higher level of training and one less experienced person with a lower level of training to represent, on average, a group of test participants as homogenous as possible. Nevertheless, because of the multitude of individual factors such as training level, experience, and physical aspects (size and strength of the emergency service personnel), it is nearly impossible to set up completely comparable teams.

The trials were designed in such a way that the same team of paramedics did not complete two runs in a row to keep the results as free as possible from exercise bias. It is worth mentioning that it would be beyond the scope of the trial to simulate a realistic operation in which emergency medical services would have to focus not only on the correct immobilization but also on other facts, such as a patient's general condition or other urgent procedures.

Furthermore, the difference between the tests with subject 1 and subject 2, as shown in the analysis of the plots summarized in S1 Appendix, gives food for thought: on the one hand, a subjectively different perception of the optimal zero angle in terms of flexion and extension may have contributed to this result. On the other hand, the differences in the stature of the two test persons may have played a role, resulting in a more or less good fit of the rigid cervical collar, which only allows limited flexibility with adjustability in three levels. This problem may be remedied by a modified study design with only one test person. Nevertheless, further biomechanical studies of the cervical spine should also take into account the aspect of population inhomogeneity and in particular the difference between the sexes, as the neck is on average less muscular in women than in men. Likewise, the aspect of differences resulting from age, such as more pronounced cervical lordosis in older people [23], has to be taken into account. A detailed consideration of the resulting forces would also be an interesting extension of the current study.

The most important point to note, however, is that both test persons were in an awake state. Although both tried to maintain an as relaxed as possible posture and avoided active movements, their handling is of course not comparable to that of patients with impaired consciousness or unconscious patients. Because such patients often experience restricted or even complete loss of muscle tone, the statements on stabilization derived from this study cannot be generally applied to patients in emergency situations.

## Conclusions

It must be determined which priority should be given to the most important rescue objective. Depending on factors such as duration of the rescue, available personnel, probability of instability of the cervical spine. as well as pain and state of consciousness of the patient, decisions must be made depending on the individual situation.

From a time point of view, use of a rescue sheet shows an advantage, from a biomechanical point of view, use of a vacuum mattress.

## Supporting information

**S1 Appendix. Plots of all 54 test runs (axial angle of rotation, lateral bending angle and angle of flexion and extension, plotted against time) for comparison.**
(PDF)

**S1 Table. Comparison of the p-values of the total time durations Ttotal for the detection of significant differences in time durations.**
(DOCX)

**S2 Table. Comparison of the p-values of the relevant parameters of the analysis of axial rotation for the detection of significant differences in these variables.**
(DOCX)

**S3 Table. Comparison of the p-values of the relevant parameters of the analysis of lateral bending for the detection of significant differences in these variables.**
(DOCX)

**S4 Table. Comparison of the p-values of the relevant parameters of the analysis of flexion and extension for the detection of significant differences in these variables.**
(DOCX)

**S1 Dataset. Complete dataset P1S0.**
(XLSX)

**S2 Dataset. Complete dataset P1S1.**
(XLSX)

**S3 Dataset. Complete dataset P2S0.**
(XLSX)

**S4 Dataset. Complete dataset P2S1.**
(XLSX)

**S5 Dataset. Complete dataset P3S0.**
(XLSX)

**S6 Dataset. Complete dataset P3S1.**
(XLSX)

**S7 Dataset. Complete dataset time markers.**
(XLSX)

**S1 File. Python scrip for transformation, smoothing and plotting.**
(DOCX)

## Author Contributions

**Conceptualization:** Sarah Morag, Martin Kieninger, Christoph Eissnert, Sebastian Dendorfer, Bärbel Kieninger.

**Data curation:** Sarah Morag, Simon Auer, Bärbel Kieninger.

**Formal analysis:** Martin Kieninger, Simon Auer, Johannes Hoffmann, Bärbel Kieninger.

**Investigation:** Sarah Morag, Martin Kieninger, Simon Auer, Bärbel Kieninger.

**Methodology:** Sarah Morag, Martin Kieninger, Christoph Eissnert, Simon Auer, Sebastian Dendorfer, Daniel Popp, Bärbel Kieninger.

**Project administration:** Martin Kieninger, Sebastian Dendorfer.

**Resources:** Martin Kieninger.

**Software:** Simon Auer, Johannes Hoffmann, Bärbel Kieninger.

**Supervision:** Martin Kieninger.

**Validation:** Simon Auer, Bärbel Kieninger.

**Visualization:** Johannes Hoffmann, Bärbel Kieninger.

**Writing – original draft:** Sarah Morag, Martin Kieninger, Bärbel Kieninger.

**Writing – review & editing:** Christoph Eissnert, Simon Auer, Sebastian Dendorfer, Daniel Popp.

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
