## [Decision Letter · Decision Letter 0]

27 Jul 2023

PONE-D-23-07268Comparison of different techniques for prehospital cervical spine immobilization: biomechanical measurements with a wireless motion capture systemPLOS ONE

Dear Dr. Kieninger,

Thank you for submitting your manuscript to PLOS ONE. After careful consideration, we feel that it has merit but does not fully meet PLOS ONE’s publication criteria as it currently stands. Therefore, we invite you to submit a revised version of the manuscript that addresses the points raised during the review process.

Three reviewers made their comments. Two clinicians recommended to accept. However, a statistical expert (reviewer 3) detected several statistical deficiencies. These deficiencies may have an impact on the results.To my mind it is an important study, which should be revised according the recommendations of reviewer 3.

We look forward to receiving your revised manuscript.

Kind regards,

Hans-Peter Simmen, M.D., Professor of Surgery

Academic Editor

PLOS ONE

“Martin Kieninger serves as academic editor for PLOS ONE.”

Reviewers' comments:

Reviewer's Responses to Questions

**Comments to the Author**

1. Is the manuscript technically sound, and do the data support the conclusions?

Reviewer #1: Yes

Reviewer #2: Yes

Reviewer #3: No

2. Has the statistical analysis been performed appropriately and rigorously? 

Reviewer #1: I Don't Know

Reviewer #2: Yes

Reviewer #3: No

3. Have the authors made all data underlying the findings in their manuscript fully available?

Reviewer #1: Yes

Reviewer #2: Yes

Reviewer #3: Yes

4. Is the manuscript presented in an intelligible fashion and written in standard English?

Reviewer #1: Yes

Reviewer #2: Yes

Reviewer #3: Yes

5. Review Comments to the Author

Reviewer #1: The authors addressed an interesting topic that is often dicussed among prehospital professionals. Very often, in recovering a severely injured, there is a"trade off" between speed and a careful recovery. The results of this study are not surprising, but objectively show, what prehospital professionals "suspect". In this sense, it's another puzzle-part that helps the responsible specialists to make decisions in the often-not-so easy prehospital setting.

Reviewer #2: Interesting biomechanical work on a currently controversial topic. The methodology was presented in detail and clearly. The results are comprehensible. In the discussion of the biomechanical data, the current controversy regarding cervical spine immobilisation is well taken into account. In contrast to many other studies, the different perspectives (rescue time, cervical spine immobilisation, etc.) are taken into account.

Reviewer #3: The outcomes of time and spinal motion were compared using three techniques (rescue sheet, vacuum mattress, long spinal board) and two conditions (with and without a rigid cervical collar). The conclusions are unclear.

Major revisions:

The study design is a two factor factorial design with repeated measures. The technique factor has 3 levels, namely rescue sheet, vacuum mattress, long spinal board, and the conditions factor has 2 levels, namely with and without a rigid cervical collar. Repeated measures were collected at several time points. Typically these types of study designs are analyzed using repeated measures two-way ANOVA if the distribution of the outcome data is normal or if it can be transformed to a normal distribution. Otherwise mixed linear regression models are used if the data does not satisfy the normality assumption of ANOVA.

Minor revisions:

1- Abstract: Briefly state the statistical methods and p-value that support the conclusions.

2- State and justify the study’s target sample size with a pre-study statistical power calculation.

The power calculation should include: (1) the estimated outcomes in each group; (2) the α (type I) error level; (3) the statistical power (or the β (type II) error level); (4) the target sample size and (5) for continuous outcomes, the standard deviation of the measurements.

3- Figure 2: Label the axes.

4- Cite the statistical software used for the analysis.

6. PLOS authors have the option to publish the peer review history of their article (what does this mean?). If published, this will include your full peer review and any attached files.

Reviewer #1: **Yes: **Stefan Matthias Mueller

Reviewer #2: No

Reviewer #3: No

---

## [Author Response · Author response to Decision Letter 0]

11 Sep 2023

Dear Prof. Dr. Simmen,

On behalf of my co-authors, I would like to thank you and the reviewers for your valuable comments and suggestions to improve the quality of our manuscript. We have revised the manuscript according to the reviewers' comments. Enclosed you find a point-by-point response to each comment indicating the action taken or the revision made.

In addition to this letter, we have uploaded a version of the revised manuscript with tracked changes ('Revised Manuscript with Track Changes') and a version without tracked changes ('Manuscript').

Thank you very much again for your help in improving the quality of our paper. We trust that our manuscript is now suitable for publication in PLOS ONE.

Sincerely,

PD Dr. Martin Kieninger

Please take note of the following statement regarding Martin Kieninger's service as Academic Editor for PLOS One: This does not alter our adherence to PLOS ONE policies on sharing data and materials.

Reviewer #3:

Major revision: 

The study design is a two factor factorial design with repeated measures. The technique factor has 3 levels, namely rescue sheet, vacuum mattress, long spinal board, and the conditions factor has 2 levels, namely with and without a rigid cervical collar. Repeated measures were collected at several time points. Typically these types of study designs are analyzed using repeated measures two-way ANOVA if the distribution of the outcome data is normal or if it can be transformed to a normal distribution. Otherwise mixed linear regression models are used if the data does not satisfy the normality assumption of ANOVA.

We have taken up the suggestion and discussed it with the biometrician at our university hospital, Dipl. Math., M. Sc. Florian Zeman. His assessment is that the approach first chosen with the simple pairwise comparison is not wrong, but still does not quite do justice to the experimental setup. Therefore, with the support of Florian Zeman, we configured an appropriate ANOVA and calculated the corresponding comparisons with our data.

Although this does not change the basic conclusions of the study, some changes in the manuscript and associated files were necessary:

In S2 Table, S3 Table, and S4 Table, the p-values were replaced by the recalculated ones.

Section Statistical analysis:

Here we now refer to the ANOVA. In addition, the content of the section had to be changed somewhat, since the Mann-Whitney U test is now only used for the evaluation of the total times, since these could not be integrated into the ANOVA. The text now reads:

Line 164-168: For a comparison of the six different experimental settings, the values (mean maximum absolute angle, mean maximum angle, and mean minimum angle) were compared using a three-way (technique, use of rigid cervical collar, time interval) ANOVA (SAF, version 9.4). A Mann-Whitney U test (2-sided, significance level 0.05) was used to compare the total times of the test runs (SPSS Statistics, version 28.0.1.0).

Section Results:

Here, all p-values had to be adjusted accordingly and, if there were changes in the significance of individual comparisons, the text had to be adjusted accordingly. The text now reads:

Line 211-221: In interval T1, both the mean maximum angle and the mean angle range of axial rotation were significantly smaller in the test setup with than in the setup without a rigid cervical collar (mean maximum absolute angle: P1S0 vs P1S1 p=<0.0001; P2S0 vs P2S1 p=0.0030; P3S0 vs P3S1 p=<0.0001; mean angle range: P1S0 vs P1S1 p=<0.0001; P2S0 vs P2S1 p=<0.0001; P3S0 vs P3S1 p<0.0001). Meanwhile, in interval T2, the mean angle range of rotation were significantly larger in the setup with a rescue sheet than in the setups with a vacuum mattress and a long spinal board (P1S0 vs P2S0 p<0.0001; P1S0 vs P3S0 p<0.0001; P1S1 vs P2S1 p=0.0189; P1S1 vs P3S1 p=0.0116). In the same interval, the mean maximum angle and the mean angle range of rotation were significantly larger in the experimental setup with a rescue sheet without a rigid cervical collar than in the same setup with a rigid cervical collar (mean maximum absolute angle: P1S0 vs P1S1 p=0.0362; mean angle range: P1S0 vs P1S1 p=0.0007).

Line 230-237: In interval T1, the mean angle range in lateral bending in all setups with a rigid cervical collar was significantly smaller than in the same setups without a rigid cervical collar (P1S0 vs P1S1 p=0.0263; P2S0 vs P2S1 p<0.0001; P3S0 vs P3S1 p<0.0001). In addition, the mean angle range is significantly smaller for the technique with vacuum mattress than for the long spinal board technique (P2S0 vs P3S0 p=0.0123; P2S1 vs P3S1 p=0.0333). Meanwhile, in interval T2, the mean angular range in lateral flexion was significantly larger for the rescue sheet technique than for the vacuum mattress and long spinal board techniques in all tests (P1S0 vs P2S0 p<0.0001; P1S0 vs P3S0 p<0.0001; P1S1 vs P2S1 p=0.0024; P1S1 vs P3S1 p=0.0105).

Line 265-276: In interval T1, the mean angle range from extension to flexion was significantly smaller in each setup with a rigid cervical collar than in the same setups without a rigid cervical collar for the rescue sheet technique and the technique with vacuum mattress (P1S0 vs P1S1 p=0.0023; P2S0 vs P2S1 p<0.0001). In interval T2, the mean minimum angle, the mean maximum angle, and the mean angle range from extension to flexion were significantly smaller in each setups without a rigid cervical collar and with the vacuum mattress and long spinal board techniques than with the rescue sheet technique (mean minimum angle: P1S0 vs P2S0 p=0.0385; P1S0 vs P3S0 p=0.0423; mean maximum angle: P1S0 vs P2S0 p<0.0001; P1S0 vs P3S0 p<0.0001; mean angle range: P1S0 vs P2S0 p<0.0001; P1S0 vs P3S0 p<0.0001). In the same interval, the mean maximum angle and the mean angle range were also significantly smaller in all tests with a rigid cervical collar in the setup with a vacuum mattress and a long spinal board than in the setup with a rescue sheet (mean maximum angle: P1S1 vs P2S1 p<0.0001; P1S1 vs P3S1 p<0.0001; mean angle range: P1S1 vs P2S1 p<0.0001; P1S1 vs P3S1 p<0.0001).

In addition, we included a note in the abstract regarding the use of an ANOVA (see next section of the response letter).

Minor revisions:

1. Abstract: Briefly state the statistical methods and p-value that support the conclusions.

We have completely revised the abstract and now provide the statistical methodology as well as the p-values:

Background

Various rescue techniques are used for the prehospital transport of trauma patients. This study compares different techniques in terms of immobilization of the cervical spine and the rescue time.

Methods

A wireless motion capture system (Xsens Technologies, Enschede, The Netherlands) was used to record motion in three-dimensional space and the rescue time in a standardized environment. Immobilization was performed by applying different techniques through different teams of trained paramedics and physicians. All tests were performed on the set course, starting with the test person lying on the floor and ending with the test person lying on an ambulance cot ready to be loaded into an ambulance. Six different settings for rescue techniques were examined: rescue sheet with/without rigid cervical collar (P1S1, P1S0), vacuum mattress and scoop stretcher with/without rigid cervical collar (P2S1, P2S0), and long spinal board with/without rigid cervical collar (P3S1, P3S0). Four time intervals were defined: the time interval in which the rigid cervical collar is applied (T0), the time interval in which the test person is positioned on rescue sheet, vacuum mattress and scoop stretcher, or long spinal board (T1), the time interval in which the test person is carried to the ambulance cot (T2), and the time interval in which the ambulance cot is rolled to the ambulance (T3). An ANOVA was performed to compare the different techniques.

Results

During the simulated extrication procedures, a rigid cervical collar provided biomechanical stability at all angles with hardly any loss of time (mean angle ranges during T1: axial rotation P1S0 vs P1S1 p<0.0001, P2S0 vs P2S1 p<0.0001, P3S0 vs P3S1 p<0.0001; lateral bending P1S0 vs P1S1 p=0.0263, P2S0 vs P2S1 p<0.0001, P3S0 vs P3S1 p<0.0001; flexion/extension P1S0 vs P1S1 p=0.0023, P2S0 vs P2S1 p<0.0001). Of the three techniques examined, the scoop stretcher and vacuum mattress were best for reducing lateral bending in the frontal plane (mean angle ranges during T1: P2S1 vs P3S1 p=0.0333; P2S0 vs P3S0 p=0.0123) as well as flexion and extension in the sagittal plane (mean angle ranges during T2: P1S1 vs P2S1 p<0.0001; P1S0 vs P2S0 p<0.0001). On the other hand, the rescue sheet was clearly superior in terms of time (total duration P1S0 vs P2S0 p<0.001, P1S1 vs P2S1 p<0.001, P1S0 vs P3S0 p<0.001, P1S1 vs P3S1 p<0.001) but was always associated with significantly larger angular ranges of the cervical spine during the procedure. Therefore, the choice of technique depends on various factors such as the rescue time, the available personnel, as well as the severity of the suspected instability.

2. State and justify the study’s target sample size with a pre-study statistical power calculation.

The power calculation should include: (1) the estimated outcomes in each group; (2) the α (type I) error level; (3) the statistical power (or the β (type II) error level); (4) the target sample size and (5) for continuous outcomes, the standard deviation of the measurements.

We were not able to perform a statistical power analysis prior to the study because we were working on a topic for which no data of this kind had been published before. This study can therefore be regarded as a pilot study with an exploratory character, in which there was no assumption about the effects. The aim was therefore to investigate whether and how the individual methods differed and to generate initial parameters for this.

The analysis showed that despite the relatively small number of replicates per setting, statistically significant results could be generated that, while not surprising to those familiar with the topic, nevertheless seem to us to be worth publishing (and Reviewer#1 and Revierwer#2 reinforce our assessment here).

Nonetheless, in response to this suggestion, we have asked ourselves whether all the statements made in the paper hold up when a subsequent power analysis is taken into account. We noticed a section where we compared the total test times with and without the rigid cervical collar and found that no statistically significant difference was detectable: if we calculated the power here, we found that it was clearly too low for a tenable statement. We have therefore adapted our manuscript as follows:

We have deleted the following sentence (line 187): The variant without a rigid cervical collar did not show any significantly shorter total duration Ttotal (P1S0 vs P1S1 p=1.000; P2S0 vs P2S1 p=0.505; P3S0 vs P3S1 p=0.436) for either of the three techniques.

Instead, we have added (line 177-178): The application of the rigid cervical collar (T0) took a mean of 22 � 8s.

We have also adjusted the following passage (line 317-318): Moreover, this additional measure did not show any appreciable prolongation of the immobilization process.

3. Figure 2: Label the axes.

We have attached the axis labels in Figure 2 and also in the figures in S1 Appendix.

4. Cite the statistical software used for the analysis.

To make our procedure even more comprehensible, we have added our python script, which we used to process the data, as another supplement (S1 Script). 

We have added a reference to the supplement in the manuscript:

Line 164: … python code S1 Script…

At the same time, we have deleted the exact name of the smoothing filter from the manuscript (line 160), since this can now be taken from the script and this information is better positioned there.

In addition, a shift in sentence components occurred in this section to better reflect the chronology of the work steps.

---

## [Editor Report · Decision Letter 1]

19 Sep 2023

Comparison of different techniques for prehospital cervical spine immobilization: biomechanical measurements with a wireless motion capture system

PONE-D-23-07268R1

Dear Dr. Kieninger,

We’re pleased to inform you that your manuscript has been judged scientifically suitable for publication and will be formally accepted for publication once it meets all outstanding technical requirements. 

Kind regards,

Hans-Peter Simmen, M.D., Professor of Surgery

Academic Editor

PLOS ONE
---

## [Editor Report · Acceptance letter]

21 Sep 2023

PONE-D-23-07268R1 

Comparison of different techniques for prehospital cervical spine immobilization: biomechanical measurements with a wireless motion capture system 

Dear Dr. Kieninger:

I'm pleased to inform you that your manuscript has been deemed suitable for publication in PLOS ONE. Congratulations! Your manuscript is now with our production department. 

Kind regards, 

on behalf of

Dr. Hans-Peter Simmen 

Academic Editor

PLOS ONE